# YOLOv8-G: An Improved YOLOv8 Model for Major Disease Detection in Dragon Fruit Stems

**DOI:** 10.3390/s24155034

**Published:** 2024-08-03

**Authors:** Luobin Huang, Mingxia Chen, Zihao Peng

**Affiliations:** 1Key Laboratory of Advanced Manufacturing and Automation Technology, Guilin University of Technology, Education Department of Guangxi Zhuang Autonomous Region, Guilin 541006, China; 2Guangxi Engineering Research Center of Intelligent Rubber Equipment, Guilin University of Technology, Guilin 541006, China; 3Guilin GLESI Scientific Technology Co., Ltd., Guilin 541004, China

**Keywords:** dragon fruit stem diseases, YOLOv8n, lightweight model, attention mechanism

## Abstract

Dragon fruit stem disease significantly affects both the quality and yield of dragon fruit. Therefore, there is an urgent need for an efficient, high-precision intelligent detection method to address the challenge of disease detection. To address the limitations of traditional methods, including slow detection and weak micro-integration capability, this paper proposes an improved YOLOv8-G algorithm. The algorithm reduces computational redundancy by introducing the C2f-Faster module. The loss function was modified to the structured intersection over union (SIoU), and the coordinate attention (CA) and content-aware reorganization feature extraction (CARAFE) modules were incorporated. These enhancements increased the model’s stability and improved its accuracy in recognizing small targets. Experimental results showed that the YOLOv8-G algorithm achieved a mean average precision (mAP) of 83.1% and mAP50:95 of 48.3%, representing improvements of 3.3% and 2.3%, respectively, compared to the original model. The model size and floating point operations per second (FLOPS) were reduced to 4.9 MB and 6.9 G, respectively, indicating reductions of 20% and 14.8%. The improved model achieves higher accuracy in disease detection while maintaining a lighter weight, serving as a valuable reference for researchers in the field of dragon fruit stem disease detection.

## 1. Introduction

According to statistics from the Ministry of Agriculture of China, the country ranks among the world’s largest producers of dragon fruit. As both domestic and international demand for dragon fruit increases, the cultivation area in China is steadily expanding [1]. Although the quality of domestic dragon fruit is relatively stable, it remains susceptible to pests and diseases such as canker, anthrax, and soft rot [2,3]. These diseases severely threaten the fruiting and ripening of dragon fruit trees, leading to significant economic losses for growers.

In the case of the dragon fruit stem disease studied in this paper, failure to administer timely treatment and appropriate pesticide application may result in the following consequences: (1) the overall health of dragon fruit plants will be compromised, hindering their normal growth and development and affecting flowering and fruiting, which will lead to a decline in fruit quality, including size, color, and taste; (2) untreated stem diseases may spread to the entire plant or even infect other healthy dragon fruit plants, potentially resulting in severe outbreaks in large dragon fruit orchards; and (3) in severe cases, stem diseases can cause the wilting or even death of dragon fruit plants, directly impacting the economic returns of growers. Therefore, timely and effective detection of stem diseases is crucial for early intervention and prevention of further disease progression and significantly mitigating the economic losses experienced by dragon fruit growers.

Currently, crop disease detection primarily relies on manual methods, where inspectors or growers visually examine the orchard and manually record the types and distribution of crop diseases. However, given the numerous types of crop diseases, including more than three stem diseases in dragon fruit alone, this process demands extensive knowledge and experience. Furthermore, varying light intensities throughout the day, especially at night, pose significant challenges for accurate disease detection by growers.

Artificial intelligence technologies, including deep learning and machine learning, have seen growing adoption and application in agriculture due to their continual advancement and widespread use [4,5]. Among these, machine learning techniques have been extensively employed for crop disease recognition. However, due to limitations in data and hardware resources, these techniques have traditionally relied on manual feature extraction and definition methods, such as histograms of oriented gradients [6], visual saliency analyses [7], and scale-invariant feature transformations [8]. These traditional techniques face several challenges, including inefficient learning, poor adaptability, and a complex feature design process, rendering them less effective in handling large volumes of image data. However, with the rapid advancements in deep learning techniques, the YOLO algorithm has achieved significant breakthroughs in image segmentation and target detection. Compared to traditional methods, YOLO offers faster detection and higher accuracy. Therefore, this paper conducts improvement research based on the YOLOv8 algorithm. The focus of the improvement is to enhance accuracy while maintaining a lighter algorithm, making it more suitable for edge detection devices.

Firstly, to achieve a more lightweight model, extensive comparative experiments on various lightweight modules were conducted. Ultimately, the C2f-Faster and CARAFE modules, which demonstrated the best overall performance, were selected for improvement. The introduction of the C2f-Faster module reduces the computational redundancy of the original C2f model and accelerates inference. Secondly, the CA mechanism and CARAFE lightweight up-sampling operator are integrated to enhance the feature extraction capability for small targets while simultaneously reducing the model size. Finally, the original model’s corrected intersection over union (CIoU) is replaced with SIoU to further enhance model stability. These improvements reduce the computational load and the number of parameters in the model, facilitating efficient and highly accurate detection of dragon fruit stem diseases. Furthermore, the lighter model is more suitable for deployment on mobile devices.

## 2. Related Work

Target detection algorithms based on deep learning are categorized into two groups: one-stage and two-stage. Two-stage algorithms operate by initially extracting candidate regions and subsequently classifying and refining the object boundaries within those regions. Due to the need for a separate network for candidate region extraction, which generates a significant number of necessary but redundant parameters during data processing, two-stage target detection and recognition algorithms, despite their high recognition accuracy, have not made breakthroughs in detection speed. Well-known algorithms such as RCNN, Fast R-CNN, and Faster R-CNN are all examples of two-stage target detection and recognition algorithms [9]. In recent years, the two-stage approach has also been applied to crop disease detection. Bari et al. [10] proposed real-time detection of rice blast, brown spot, and hispa diseases using Faster R-CNN, achieving an mAP of 98.7%. Afzaal et al. [11] addressed the scarcity of strawberry disease datasets by creating a dataset containing 2500 images. They applied the Mask R-CNN algorithm to this dataset and employed data augmentation techniques to segment instances of strawberry diseases, achieving an mAP50 of 82.43%.

Although two-stage algorithms achieve high detection accuracy, their slow processing and heavy models render them unsuitable for deployment on mobile devices. In contrast, the YOLO series, a one-stage detection algorithm, offers significant advantages in detection speed and relatively low model weights, making it more suitable for mobile device deployment.

The YOLO series is a one-stage detection algorithm that has a great advantage in detection speed [12,13]. The core idea of the algorithm is to use the whole map as the input to the network. The algorithm combines the two steps of target determination and target identification into a single step. This algorithm directly regresses the position coordinates and classification probability of the bounding box, merging the localization and classification steps and requiring only a single scan of the image. The neural network performs feature extraction, classification, and localization in one pass, significantly enhancing the speed of target detection. YOLO employs a single convolutional neural network for target detection, i.e., its training and prediction are end to end. Its straightforward network structure aligns more closely with the principles of deep neural networks compared to two-stage detection algorithms. YOLO incorporates measures to prevent overfitting, resulting in a more robust model. Additionally, its light weight facilitates transfer learning. The YOLO algorithm has been developed into several series. In YOLOv1 [14], the authors introduced a novel loss function that integrates the bounding box coordinates with the confidence level of category predictions to form a comprehensive regression loss function. Furthermore, YOLOv1 employs multi-scale training and data augmentation techniques to enhance detection performance. YOLOv2 [15,16] improves upon YOLOv1 by proposing a multi-scale prediction method and an anchor box mechanism. The multi-scale prediction method allows for predictions on feature maps of varying scales, enhancing detection performance. The anchor box mechanism generates a set of representative anchor boxes through clustering the training data, enabling the model to better adapt to targets of different sizes. YOLOv3 [17] further optimizes the network structure and training strategy by incorporating residual connections, multi-scale predictions, and cross-layer connectivity. These advancements enhance both detection speed and accuracy. YOLOv5 [18] adopts a variety of acceleration strategies, such as the use of CSP structure, streamlining the network structure, and the use of TensorRT, etc., which gives it a great advantage in terms of speed. YOLOv8 replaces the C3 module with the C2f module, leading to a reduction in model parameters. Additionally, YOLOv8 employs the anchor-free detection approach, which enhances its speed while maintaining accuracy comparable to other target detection methods. One-stage algorithms are not limited to YOLO: SSDs are also one-stage algorithms widely used in detection tasks. Sun et al. [19] proposed MEAN block by reconstructing the convolution module, introduced the inception module of GoogLeNet, replaced all the convolutions in it with the MEAN block module, and proposed a mobile-based SSD algorithm for the detection of assessed leaf diseases through the combination of MEAN block and Inception. Even though the detection speed has improved over Faster-RCNN, it still falls short of the mobile terminal’s current detection requirements. Therefore, researchers have begun utilizing the YOLO algorithm for crop disease detection. Liu et al. [20] optimized the feature layer for YOLOv3 using an image pyramid for multi-scale detection of different tomato disease sizes and improved the quality of tomato disease detection. Xue et al. [21] addressing the challenges of shadow, overexposure, and small spots in leaf images of tea disease, introduced the attention mechanism into the YOLOv5 model. They replaced the pooling layer with the receptive field block (RFB) and proposed the YOLOv5-tea network model. Compared to classical algorithms and the original model, YOLOv5-tea effectively improves the detection of shadowed and overexposed images. To enhance the robustness of the detection model. Hu et al. [22] enhanced the dataset by employing data augmentation techniques such as mosaic, cutout, and mix-up. These methods expanded the tomato fruit dataset and increased the model’s generalizability, enabling the detection model to adapt to varying orchard environments for tomato fruit detection. Since crop diseases can manifest in various sizes and shapes, Liu et al. [23] utilized the 2D deformable convolution (DCNv2) module to significantly enhance the detection of cucumber early-blight leaves of varying sizes and shapes. By modifying the loss function, the model’s accuracy was further improved. Although these algorithms have achieved faster detection, they still have relatively large models. To further minimize the size of the YOLO model, Yang et al. [24] lightened the neck network of YOLOv8 by introducing slim-neck modules, which simplified the model structure and shortened the time of spot detection. Cheng et al. [25] proposed the GC2f module by replacing the traditional convolution in C2f with the lightweight GhostConv, which optimizes the structure of C2f in the backbone and neck networks and reduces the number of parameters in C2f, resulting in a reduction in the overall size of the model, which is more conducive to deployment on embedded devices. Boudaa et al. [26] conducted detection studies on crop diseases affecting apples, cucumbers, grapes, and tomatoes using YOLOv5s, YOLOv8s, and YOLOv9-c, respectively. The experimental results indicated that YOLOv9-c achieved the best detection performance with an accuracy of 93.1%, 1.3% and 42.5% higher than YOLOv8s and YOLOv5s, respectively. However, FLOPS was not considered, with YOLOv9-c reaching 102.1 G compared to YOLOv8s’ 28.6 G. The high computational demand of the YOLOv9-c model poses a significant challenge for deployment on small devices.

In summary, the YOLO algorithm has been widely applied to various major crops and has demonstrated high effectiveness in crop disease detection. However, its application to dragon fruit stem disease detection is limited. Therefore, this paper proposes an improved YOLOv8-G algorithm specifically for detecting dragon fruit stem disease, optimized for model size and computational load to make the improved model lightweight, meeting the deployment requirements of edge devices. These improvements reduce the computational load and number of parameters in the model, enabling efficient and highly accurate dragon fruit stem disease detection.

## 3. Methodology and Design

### 3.1. YOLOv8 Model

YOLOv8 is an advanced detection technique that is a version of the YOLO series of algorithms. It is specifically created for tasks like identifying targets, classifying images, and segmenting instances. The YOLOv8 model structure is depicted in Figure 1. The backbone, neck, and head are components of the YOLOv8 algorithm [27].

The backbone module is used for the extraction of features from images. The system is composed of convolutional (Conv), C2f, and spatial pyramid pooling with factorized convolution (SPPF) modules. The Conv component utilizes convolution kernels to extract features while minimizing the loss of gradient flow information. The C2f module in YOLOv8 improves upon the ELAN concept in YOLOv7 by integrating the idea of gradient flow. This improvement is based on the combination of ideas from YOLOv5 and YOLOv7, resulting in a better C2f module [28]. The C2f structure is utilized by the YOLOv8 model to obtain more gradient flow information and improve its feature extraction capability. The C2f structure is depicted in Figure 2. Additionally, the SPPF module, which is responsible for spatial pyramid pooling, is used to convert feature maps of varying sizes into a fixed-size feature vector. Finally, feature splicing is performed.

The neck section of the model is designed based on the feature pyramid network (FPN) [29] and the path aggregation network (PAN) [30]. This design effectively integrates top-down and bottom-up information flows within the network, thereby expanding the receptive field and incorporating multi-scale information into the feature map. Consequently, it enhances the overall detection performance.

The head module employs feature maps of varying sizes to extract both category and location information for objects of varied scales. The distributed focal loss (DFL) concept [31] is implemented to decrease the parameter size and computational complexity. In-stead of using anchor-based, anchor-free is used, which characterizes the object by the boundary information of multiple key points, midpoints, and boundaries. This approach enhances the accuracy of object localization and is particularly well-suited for dense detection.

The training and prediction process of the YOLOv8 model consists of the following main steps.

Model training: (1) Pass the input image through the YOLOv8 model to generate the predicted bounding box and category probabilities. (2) Calculate the loss function based on the predicted values and ground truth labels. (3) Calculate the gradient and update the model parameters by using the using the backpropagation algorithm. (4) Evaluate the model performance on the validation set and adjust the hyperparameters to obtain the best results.

Model prediction: (1) Load the trained YOLOv8 model weights. (2) Preprocess the input image, such as by adjusting the image size and normalizing the data. (3) Input the preprocessed image into the YOLOv8 model for forward propagation to generate the predicted bounding boxes and category probabilities. (4) Postprocess the model output, including non-maximal suppression (NMS), to remove overlapping bounding boxes and retain optimal detection results. (5) The predicted results are visualized in the image.

The YOLOv8 series has excellent performance in both detection and response time. In this study, YOLOv8 is used as the basis for improvement, based on which the model size, detection accuracy, and computational load are further optimized with the characteristics of dragon fruit stem disease. While enhancing detection performance, the FLOPS are reduced to make the model more suitable for edge device detection.

### 3.2. Introduction to the Improved YOLOv8 Network

The fundamental structure of YOLOv8 remains unchanged. However, certain components have been enhanced or substituted. The YOLOv8-G model is introduced, and its description is provided in this section. The network architecture of YOLOv8-G is illustrated in Figure 3.

#### 3.2.1. Feature Extraction Module of C2f-Faster

In actual dragon fruit plantations, the captured images often have cluttered backgrounds with weeds, insects, and other interferences, and the diseased areas are frequently obscured by other dragon fruit stems. In such cases, convolutional neural networks (CNNs) often introduce redundant features and noise during image feature extraction, which impedes the model’s ability to accurately identify dragon fruit stem diseases. Each convolutional operation adds complexity to the model, necessitating additional computational resources [32,33].

Eliminating interfering feature redundancy reduces model complexity, improving target detection accuracy. MobileNets, ShuffleNets, and GhostNet, along with other models, utilize depthwise convolution (DWConv) and group convolution (GConv) techniques to extract spatial data. Nevertheless, when attempting to decrease FLOPS, operators often experience the drawback of heightened memory access. FasterNet [34] introduced the concept of partial convolution (PConv), which leverages redundancy in feature mapping by selectively performing regular convolution on specific input channels while leaving others unaffected. Additionally, a pointwise convolution (PWConv) is integrated into PConv to maximize information utilization across all channels. Every FasterNet Block is equipped with a PConv layer that is accompanied by two PWConvs, or 1 × 1 Conv layers positioned directly behind it. Together, they form the structure of an inverted residual block, characterized by an expanded number of channels in the middle layer and the addition of shortcuts to reuse input features. FasterNet, which is built using the FasterNet Block, exhibits excellent performance in both classification and detection tasks. This study enhances the C2f module in the YOLOv8 network by integrating the FasterNet Block, resulting in a new structure termed C2f-Faster. This structure is employed for feature extraction and processing in the detection of dragon fruit stem disease. By applying this approach, the quantity of parameters and computational complexity are reduced while simultaneously preserving a wide range of sensory fields and nonlinear representation capabilities. The network structure of C2f and C2f-Faster is depicted in Figure 4.

#### 3.2.2. CARAFE Lightweight Operator

The CARAFE up-sampling operator stands out for its lightweight design and adapt-ability to features of varying content and size [35]. Guided by the semantic information inherent in the input feature map, it can direct the formation of the up-sampling kernel, extending the receptive field and enhancing the overall quality of the up-sampling process. For the dragon fruit stem disease dataset, this enhances the model’s feature extraction from low-resolution images, leading to better detection outcomes. This versatile up-sampling operator can predict and adjust the up-sampling kernel using the input feature map to improve the up-sampling process.

The up-sampling technique in YOLOv8 employs standard interpolation, prioritizing the spatial details of the input feature map while disregarding its semantic information. In the detection of dragon fruit stem diseases, the use of standard interpolation can result in information loss, ambiguity, a limited receptive field, and decreased detection efficiency [36,37]. Therefore, this study presents the CARAFE lightweight up-sampling operator as a solution to these challenges. The workflow diagram of the CARAFE up-sampling operator is displayed in Figure 5.

CARAFE consists of two blocks: the up-sampling kernel prediction module and the feature reorganization module. Firstly, when provided with an input feature map size of C×H×W, the information channel is compressed to $H\times W\times C_{m}$ by a 1×1 convolution. This significantly reduces the computational workload for subsequent processes. After compressing the information channel, it is processed through a convolutional layer. This layer predicts the up-sampling kernel, which modifies the input feature map. The kernel is then expanded in the spatial dimension to produce the final up-sampled kernel, denoted $\sigma H\times \sigma W\times K_{up}^{2}$. Additionally, each channel of the kernel is normalized using the softmax function to guarantee the consistency of its weights. In the feature reconstruction stage, each position in the output feature map is correlated with the corresponding region in the input feature map. This is achieved by extracting the central region $K_{up}\times K_{up}$ and performing a dot product with the expected up-sampling kernel at that location to generate the output value. Channels at the same location in C×σH×σW have a shared up-sampling kernel, resulting in the production of the output feature map.

#### 3.2.3. CA Attention Mechanism

The attention mechanism is an efficient way of allocating information resources based on the characteristics of human attention. This method enhances the focus of deep learning neural networks on areas that require attention. It significantly improves neural network performance and is widely used in computer visual detection tasks [38]. Nevertheless, the conventional attention mechanism often overlooks the spatial location information of the target [39]. Instead, CA considers the relationship between the channel information and the orientation-related location information, thereby assisting the model in accurately locating and identifying disease features. The process of implementing CA attention is as follows.

(1)To begin with, the input layer has a size of C×H×W, with each channel encoded horizontally and vertically using pooling kernels of size (H,1) and (1,W), respectively. The outputs of the cth channel, with a height of H and width of W, are represented by Equations (1) and (2) correspondingly:

(1)$$Z_{c}^{h}(h)=\frac{1}{W}\sum_{0\leq i<w}x_{c}(h,i)$$(2)$$Z_{c}^{w}(w)=\frac{1}{H}\sum_{0\leq i<H}x_{c}(j,w)$$The variables H and W represent the height and width of the feature layer, respectively. The variables $Z_{c}^{h}$ and $Z_{c}^{w}$ represent the outputs of the cth channel along the H and W directions, separately. The variables $x_{c}(h,j)$ and $x_{c}(j,w)$ represent the inputs of the input feature maps X along the H and W directions.

(2)The process of CA creation involves combining the feature maps from two independent directions using a 1×1 convolution operation and an activation operation, as depicted in Equation (3).

(3)$$f=\delta (F_{1}([Z^{h},Z^{w}]))$$The symbol f represents the intermediate feature map obtained by encoding the spatial information in both the horizontal and vertical directions. The operation [·, ·] symbolizes the concatenation of feature maps along the spatial dimension. The symbol $F_{1}$ represents the convolution operation, while δ represents the sigmoid function.

The feature map f is segmented along the spatial dimension. Two 1×1 convolution operations are subsequently applied to modify the number of channels in f, aligning it with the number of channels in the input feature X. Ultimately, the sigmoid function is employed to compute the attention weight of the feature map. The findings are derived from Equations (4) and (5).
(4)$$g^{h}=\sigma (F_{h}(f^{h}))$$
(5)$$g^{w}=\sigma (F_{w}(f^{w}))$$The variables $g^{h}$ and $g^{w}$ represent the attentional weights for convolution and activation after splitting for f. Similarly, $f^{h}$ and $f^{w}$ represent the feature maps along the H and W directions, respectively, after splitting for f. Lastly, $F_{h}$ and $F_{w}$ designate the convolution operations.

(3)Compute the product of $g^{h}$ and $g^{w}$ with the weights of the input feature map to generate the feature map, where the weights are determined by the coordinate attention weights described in Equation (6):

(6)$$y_{c}(i,j)=x_{c}(i,j)\cdot g_{c}^{h}(i)\cdot g_{c}^{w}(j)$$The variables $y_{c}(i,j)$ and $x_{c}(i,j)$ represent the input and output of the cth channel, respectively. The variables $g_{c}^{h}(i)$ and $g_{c}^{w}(j)$ represent the attentional weights on the cth channel along the H and W directions, separately.

The CA applies horizontal and vertical attention weights to weight the input feature maps horizontally and vertically to more accurately locate the exact position of the object of interest, which in turn helps the whole model to better recognize the object. This can reduce the number of missed detections for small targets such as canker and anthrax. The workflow of the CA is shown in Figure 6.

#### 3.2.4. SIoU Loss Function

The loss function serves as a metric for the discrepancy between the predicted information and the desired information. It also controls the positioning of the recognition box, thereby enhancing the accuracy of the prediction outcome [40]. The loss function of the YOLOv8 network consists of three components: deep feature loss, classification loss, and localization loss [41,42]. Initially, the regression loss is calculated using IoU, which measures the overlap between the predicted box and the ground truth box. The IoU is computed as follows:(7)$$IoU=\frac{\left|A\cap B\right|}{\left|A\cup B\right|}$$

In Equation (7), A represents the predicted box, B represents the ground truth box, A∩B denotes the intersection region between the predicted box and the ground truth box, and A∪B represents the merging area between the predicted box and the ground truth box.

However, if there is no overlap between the predicted box and the ground truth box, the IoU value will be zero, as defined in Equation (7). This means that the distance between the two boxes cannot be determined, resulting in a gradient of zero and hindering optimization. To address this issue, this paper employs the SIoU loss function [43]. SIoU is a loss function used to evaluate the degree of overlap between predicted and true bounding boxes in target detection algorithms. It improves upon the traditional IoU by further considering the vector angle between the real and predicted boxes, thereby redefining the associated loss function.

The SIoU loss function comprises four components: angular loss, distance loss, shape loss, and IoU loss.

The angular loss $L_{angle}$ process is shown in Figure 7a. The center B $\left(b_{c_{x}},b_{c_{y}}\right)$ of the predicted box is taken as the starting point.

The transverse or longitudinal axes are set to the distance from the center of the ground truth box B^GT^ $\left(b_{c_{x}}^{gt},b_{c_{y}}^{gt}\right)$ to reduce the degrees of freedom of the anchor frame, thus enabling rapid convergence to the ground truth box along the *x*-axis or *y*-axis. The variables σ and $c_{h}$ denote the distance and height difference between the center point of the ground truth box and that of the predicted box, respectively, as calculated in Equation (8).
(8)$$L_{angle}=1-2\times \sin^{2}(\arcsin(\frac{c_{h}}{\sigma })-\frac{\pi }{4})=1-2\times \sin^{2}(\alpha -\frac{\pi }{4})$$

The distance loss $L_{dis}$ process is shown in Figure 7b, and the diagonal distance between the prediction box and the minimum outer rectangle of the real box is calculated as shown in Equation (9), where $c_{w}$ and $c_{h}$ represent the width and height of the minimum outer rectangle.
(9)$$\left\{\begin{array}{c} L_{dis}=2-e^{-\gamma \rho_{x}}-e^{-\gamma \rho_{y}}\\ \rho_{x}={\left(\frac{b_{c_{x}}^{gt}-b_{c_{x}}}{c_{w}}\right)}^{2}\\ \rho_{y}={\left(\frac{b_{c_{y}}^{gt}-b_{c_{y}}}{c_{h}}\right)}^{2}\\ \gamma =2-\Lambda \end{array}\right.$$

The shape loss $L_{shape}$ is calculated as shown in Equation (10), where w, h, $w^{gt}$, $h^{gt}$ are the width and height of the predicted box and the ground truth box, respectively, and θ controls the degree of attention to the shape loss so as to avoid paying too much attention to the shape loss of the dragon fruit stem lesions and reducing the movement of the predicted box during detection.
(10)$$\left\{\begin{array}{c} L_{shape}={\left(1-e^{-w_{w}}\right)}^{\theta }+{\left(1-e^{-w_{h}}\right)}^{\theta }\\ w_{w}=\frac{\left|w-w^{gt}\right|}{\max(w,w^{gt})}\\ w_{h}=\frac{\left|h-h^{gt}\right|}{\max(h,h^{gt})} \end{array}\right.$$

The formula for obtaining the SIoU loss function is shown in Equation (11), where IoU is the correlation between the true defect location and the prediction.
(11)$$L_{SIoU}=1-IoU+\frac{L_{dis}+L_{shape}}{2}$$

The SIoU optimizes the predicted box by ensuring that the centroids of the predicted box converge to the *x*-axis. More precisely, the centroids of the predicted box and the ground truth box are modified to align with the *x*-axis. Additionally, the distance between the two centroids in the *x*-axis direction is decreased even more when the two boxes have a similar shape. In this procedure, the width and height of the anticipated box, together with the distance between the two center points in the *y*-axis direction, are consistently modified to align the predicted box with the ground truth box as closely as possible [44].

As a result, aligning the predicted box with the *x*-axis of the ground truth box significantly enhances the model’s regression efficiency and accelerates the convergence of the loss function. Simultaneously, converging the predicted box along the diagonal of the ground truth box can reduce the center coordinates of the predicted box on the x and y axes by a certain proportion, hence enhancing the model’s optimization.

By incorporating the target’s geometric structure information, SIoU enhances the accuracy of regression prediction by more precisely matching the target object and reducing bounding box errors. Additionally, SIoU mitigates uncertainty and fluctuations during the bounding box regression process, thereby improving the stability and robustness of the model across various scenarios.

## 4. Experimental Results and Analysis

### 4.1. Experimental Dataset

#### 4.1.1. Dataset Preparation

This study primarily focused on detecting major diseases affecting dragon fruit stems, specifically canker, anthrax, and soft rot. Information on the features of these three diseases, along with associated photos, is provided in Table 1. The acquisition of the dataset was split into two parts. One part consisted of image data obtained from the Google Dataset Search public database. The other part consisted of images captured on-site at the dragon fruit park of the Guangxi Academy of Agricultural Sciences. The filming was conducted using a Redmi K60 mobile phone, and the filming distance was controlled within a range of 10 to 40 cm [45]. The collection comprised photos depicting the three distinct stages of the diseases: early, intermediate, and late. To ensure the accuracy of the dataset, the types of dragon fruit stem diseases were identified through online academic resources and consultations with fruit producers and experts.

#### 4.1.2. Data Preprocessing

Once the data collection of dragon fruit stem diseases images is completed, the first step is screening and cropping. Images with blurred and large light spots in the diseased portion are eliminated, and images with a moderate size of disease are cropped. A Python script adjusts the image resolution to 640 × 640 in the second step, followed by data enhancement in the third step. The enhancement methods include image rotation, brightness adjustment, scaling, and Gaussian noise addition. Image rotation and scaling simulate various angles and distances during detection, while adjusting brightness and adding Gaussian noise mimic different external lighting conditions and mosquito interference in the orchard. These data augmentation techniques effectively simulate diverse external environments [46,47]. The data augmentation methods simulate various external environments, enhancing the diversity and complexity of the dataset. This approach improves the usability and robustness of the disease detection model. The improved picture data is displayed in Figure 8 below. The count of original photographs and the count of images after data augmentation are shown in Table 2. The dataset was separated, allocating 80% for training and 20% for testing. This ratio ensures that the model receives sufficient samples for learning, allowing it to adequately capture the disease features in the dataset. Additionally, it provides an independent evaluation criterion to help detect potential overfitting issues.

### 4.2. Test Platform and Parameter Settings

The training and testing of dragon fruit stem disease images were conducted using the PyTorch framework, and the configuration of the experimental environment is shown in Table 3. The hyperparameters for training in this study used the default values of YOLOv8, and the same hyperparameters were used for all other comparison experiments. The training hyperparameters are set as shown in Table 4 below.

### 4.3. Indicators for Model Evaluation

This study employs precision, recall, and mAP to assess the accuracy of the disease detection model for dragon fruit stem. The objective is to substantiate the functionality of the YOLOv8-G model. The following are the formulas for mAP, recall, and precision:(12)$$P=\frac{TP}{TP+FP}$$
(13)$$R=\frac{TP}{TP+FN}$$
(14)$$AP=\int_{0}^{1}P(R)dR$$
(15)$$mAP=\frac{\sum_{q=1}^{Q}AP(q)}{Q}$$

The variables TP, FP, and FN represent the count of properly predicted, erroneously predicted, and missing dragon fruit stem disease targets, respectively. The variable q represents the number of detection categories, which is set to three in this research.

Additionally, this study evaluates the lightweight characteristics of the model using the following parameters: model size, FLOPS, number of parameters, and frames per second (FPS). In this context, FLOPS quantify the floating-point operations performed by the model during inference, while parameters represent the number of model parameters. Smaller values for either FLOPS or parameters indicate a lower level of algorithmic complexity or model. Additionally, FPS denotes the number of frames of the detected image per second, serving as a metric for measuring the speed of detection. The model size refers to the weight of the inference model. A smaller weight is better suited for deploying the detection model in practice.

### 4.4. Comprehensive Comparative Analysis

#### 4.4.1. Comparison before and after Improvement

(1)Comparison of model performance before and after improvement

To assess the impact of the model’s improvement, both the original and improved models were directly analyzed using evaluation indexes. The experimental results of this study were obtained from the testing set. The experimental results are presented in Table 5, indicating that the precision, recall, mAP50, and mAP50:95 of the improved YOLOv8-G model increased by 1%, 5%, 3.3%, and 2.3%, respectively. The visualization curves are displayed in Figure 9. The dimensions of the enhanced model are just 4.9 MB. The FLOPS are 6.9 G. Compared to the original YOLOv8n model, the size of the model is reduced by 20%, and the FLOPS are reduced by 1.2 G. This suggests that the improved model can achieve better detection of dragon fruit stem disease while requiring fewer computational resources. It improves the overall performance of the model and makes it more suitable for deployment on edge devices for detection purposes.

(2)Comparison of disease detection effects before and after model improvement

A comparison of the detection performance before and after the improvement of YOLOv8 is presented in Figure 10. Figure 10 shows that the improved algorithm has increased the confidence level in identifying various types of dragon fruit stem diseases compared to the original model. The enhanced model more efficiently detects small target lesions, ensuring higher detection rates and minimizing the risk of missed detections. In brief, the enhanced YOLOv8-G algorithm demonstrates the capability to accurately predict small lesions with a high level of certainty. Additionally, it exhibits improved confidence in detecting lesions of regular size, making it more versatile and suitable for the implementation of dragon fruit stem disease detection devices.

#### 4.4.2. Comparison of Different Detection Models

Various target detection models, including Faster-RCNN, SSD, YOLOX, and YOLOv8, were employed to train the dragon fruit stem disease dataset. These models were compared with the YOLOv8-G model introduced in this study. The experimental results are presented in Table 6. These show that YOLOv8n achieves a precision of 83.2% and a recall of 75.2%. The mAP50 of YOLOv8n was 79.8%, which was 24%, 1.7%, and 1.2% higher than the other detection models except YOLOv9-t and YOLOv8-G, respectively. Furthermore, the YOLOv8n model is considerably smaller compared to other models, measuring only 6 MB. This characteristic provides a notable benefit for its deployment on mobile devices, thereby confirming the validity of the paper’s improvement based on YOLOv8n. YOLOX-tiny, a lighter model from the YOLOX series, underperforms compared to YOLOv8n in both object recognition accuracy and model size. In this study, the Faster-RCNN and SSD models achieved mAP50 scores of 55.8% and 78.1%, respectively. Furthermore, due to their relatively large models, they are not suitable for the lightweight requirements of mobile devices. While YOLOv9-t’s mAP is 2% higher than that of YOLOv8n, it also incurs a 26% increase in FLOPS, indicating a larger computational load. On the other hand, the improved YOLOv8-G provides multiple benefits regarding mAP50 and model size. It shows significant improvements of 27.3%, 5%, 4.5%, and 3.3% in mAP50 and 16.6%, 14.4%, 10.1%, and 5% in recall when compared to Faster-RCNN, SSD, YOLOX, and YOLOv8n, respectively. Additionally, the model size is only 4.9 MB, which is 1.1 MB smaller than YOLOv8n. Furthermore, the detection speed reaches 95.2 frames·s^−1^, meeting the real-time detection requirement of edge devices. The YOLOv8-G model suggested in this study outperforms existing models in terms of both model size and computation. Additionally, it exhibits superior detection accuracy and overall performance compared to other models.

#### 4.4.3. Comparative Experiments of Attention Mechanisms

To verify the effectiveness of the attention mechanism proposed in this study, various attention modules were integrated into the same location within the model structure. These modules included the convolutional block attention module (CBAM), the squeeze-and-excitation (SE) module, and the simultaneous attention module (SimAM). Each module was added after the backbone network’s SPPF layer. The experimental results are displayed in Table 7. From Table 7, it is evident that the addition of the attention mechanism results in only a slight increase in the number of model parameters. However, the mAP50 shows a relatively obvious improvement compared to the original model. Among the attention mechanisms, the CA has the most pronounced effect on model accuracy, enhancing it by 2.3%.

#### 4.4.4. Comparison of Different Loss Functions

During model training, the loss function is responsible for backpropagation, which adjusts the network weights and learned features through iterative learning to minimize prediction errors. The commonly used loss functions include CIoU, DIoU, EIoU, GIoU, and SIoU. The YOLOv8n base model uses CIoU as the loss function. To evaluate the effect of different loss functions on the mAP50 of disease detection, the CIoU in the original model was replaced with the above loss function for comparative testing. Additionally, focal loss was introduced for comparison. The comparison results are presented in Table 8. The results indicate that the YOLOv8n model achieves the highest mAP50 of 82.2% when the loss function is replaced with SIoU. Additionally, the introduction of focal loss further improves the accuracy by 0.2%, resulting in a total improvement of 2.6% compared to the original YOLOv8n model. This suggests that using the focal SIoU loss function improves the detection performance for datasets with large differences in target size.

#### 4.4.5. Ablation Test

To validate the efficacy of the newly incorporated module, an ablation test is conducted on C2f-Faster, CA, CARAFE, and SIoU. The experimental results, as shown in Table 9, will be compared and assessed based on detection accuracy and the number of model parameters. Table 9 demonstrates that the introduction of the FasterNet lightweight network in the C2f module enhances the processing speed of the model and decreases the number of parameters. This suggests that the convolution module of the original model is computationally redundant, leading to an excess of unnecessary parameters and channels. Following the implementation of the CA, the model’s mAP50 and recall experienced a noticeable enhancement. This improvement can be attributed to the CA attention mechanism’s ability to enhance feature extraction and expand the receptive field. The results of the experiment showed that the mAP50 increased by 0.5%. Despite a slight increase in model size, less than 0.1 MB, the model’s detection performance was not compromised. By replacing the up-sampling operator with CARAFE, the mAP50 increased by 1.3%, accompanied by a slight rise of 0.4 G in FLOPS. The CARAFE up-sampling operator enhances semantic information understanding within low-resolution feature maps. Through feature reorganization, it improves the model’s accuracy in detecting small target diseases. By replacing CIoU with SIoU and integrating focal loss, the issue of directional mismatch between the predicted box and ground truth box was resolved. This resulted in accelerated model convergence and a 0.8% improvement in mAP50. A thorough comparison reveals that YOLOv8-G, when compared to the original YOLOv8n model, not only enhances the mAP50 and recall rate of dragon fruit stem disease detection by 3.3% and 5%, respectively, but also reduces the model size by 20% and the FLOPS by 16%. Overall, the YOLOv8-G model offers enhanced accuracy and reduced complexity, making it ideal for precise and effective detection of dragon fruit stem disease. Its optimized model size and FLOPS align well with the deployment needs of mobile devices.

## 5. Conclusions

This paper proposes an enhanced algorithm, YOLOv8-G, which achieves higher detection accuracy compared to the original model, reduces the incidence of missed disease detections, and features a lighter model architecture. Based on the conducted experiments, the following findings have been derived.

(1)FasterNet is integrated into the C2f module of the backbone and neck networks to form the C2f-Faster module. This modification produces a lighter model with enhanced inference speed and reduced computational parameters.(2)The incorporation of CA into both the backbone and neck networks significantly enhances the model’s ability to accurately identify and extract features.(3)Employ the SIoU as a substitute for the CIoU and incorporate the classification focal loss to enhance the model’s accuracy, reduce the loss of image information during backpropagation, and enhance the stability of the model.(4)The substitution of the up-sampling operator with CARAFE enhances feature extraction from low-resolution images through its aggregation step. This process reorganizes features to generate a high-resolution composite image, thereby improving the accuracy of detecting small target diseases in low-resolution scenarios.

It can be concluded that the improved YOLOv8-G model based on YOLOv8 outperforms other models, achieving an FPS of 87.3 frames per second with a model size of only 4.9 MB, which is 20% lighter than the original model. The FLOPS of the model are also reduced to only 6.9 G. Furthermore, the enhanced YOLOv8 model has an mAP50 of 83.1%, surpassing YOLOv8n by 3.3%.

In practical agricultural production applications, a lightweight model is advantageous for small, low-performance edge devices. This reduces the hardware requirements for inspection devices, enabling the use of more cost-effective hardware and subsequently lowering production costs. By applying the data enhancement method, the dataset in this study becomes more diversified, enhancing the robustness and practicality of the proposed model. In summary, the improved YOLOv8-G algorithm is more effective for detecting stem diseases in dragon fruit and is relatively lightweight, making it highly significant for the promotion and implementation of smart agriculture.

## 6. Discussion

Although the improved YOLOv8-G model proposed in this paper is 20% lighter than the original model and shows improved detection accuracy, it still cannot detect all the diseased regions of dragon fruit stems. For some tiny diseases and cases with occlusion between diseased areas, missed and false detections persist. For example, as illustrated in Figure 11, the model proposed in this paper does not perform well in detecting some regions with tiny disease spots and shaded areas.

Additionally, there remains potential for further model size reduction. Future research will aim to enhance the activation function, incorporate other network structures, and explore the use of TensorRT to achieve faster inference and a smaller model size. Experimentally, the model will be deployed on an actual machine vision development board equipped with an industrial camera for real-time detection of dragon fruit stem diseases in the orchard. Furthermore, corresponding dragon fruit stem disease detection software will be developed for mobile devices, enabling the trained model to be imported into the software for on-the-go disease detection.

## Figures and Tables

**Figure 1 sensors-24-05034-f001:**
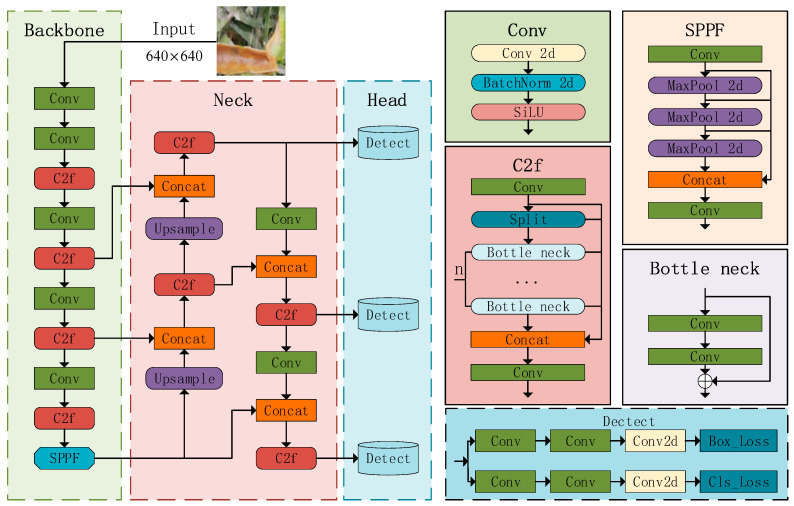
The network architecture of the YOLOv8.

**Figure 2 sensors-24-05034-f002:**
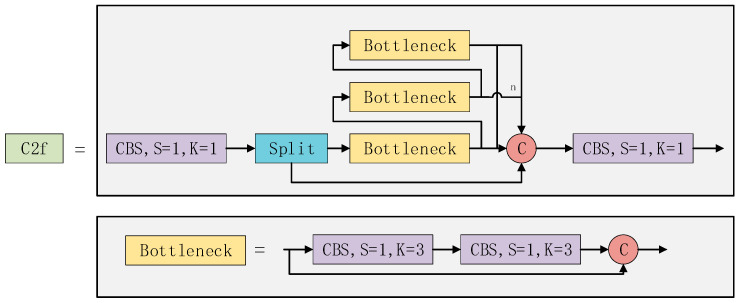
The internal structure of the C2f module.

**Figure 3 sensors-24-05034-f003:**
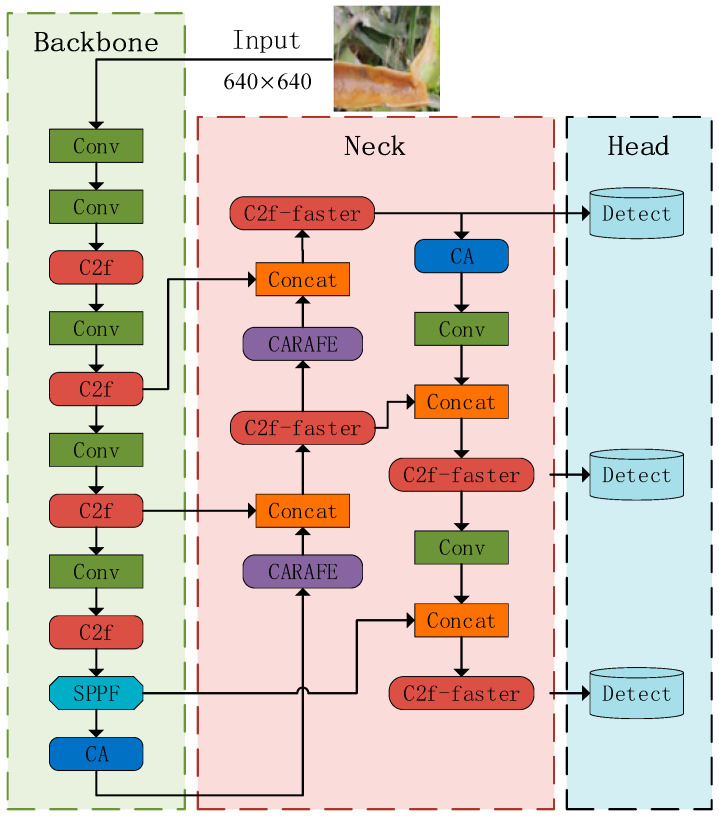
The network architecture of the improved YOLOv8-G.

**Figure 4 sensors-24-05034-f004:**
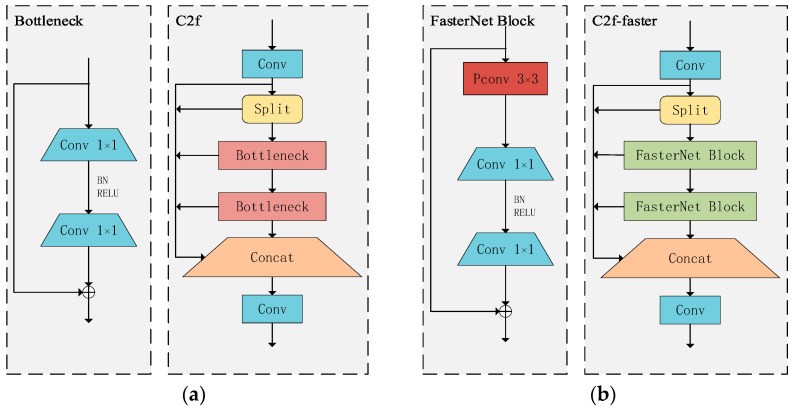
Comparison of C2f modules and C2f-Faster modules: (**a**) network structure of C2f; (**b**) network structure of C2f-Faster.

**Figure 5 sensors-24-05034-f005:**
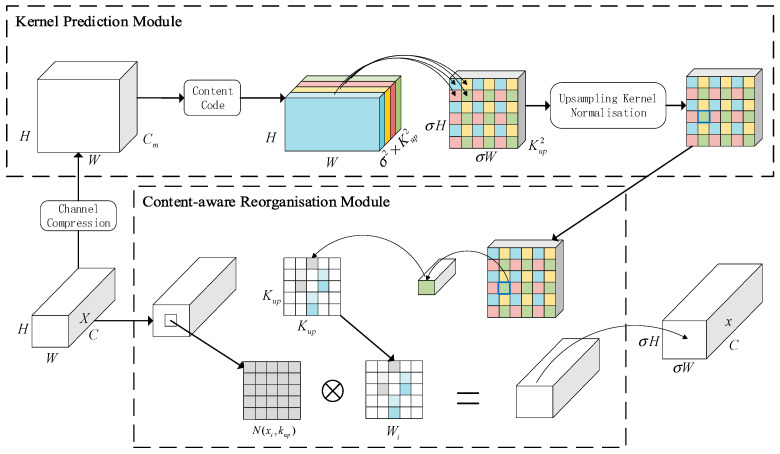
Flowchart of the CARAFE.

**Figure 6 sensors-24-05034-f006:**
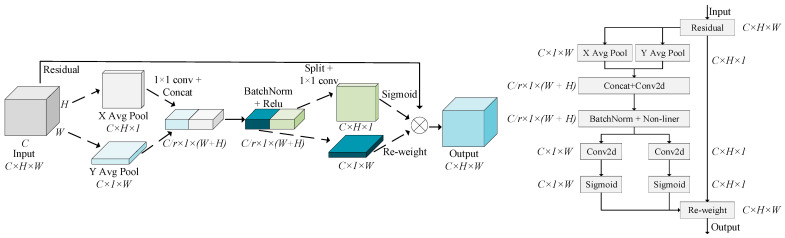
Structure diagram of the CA attention mechanism. Note: C, H, and W denote the number of channels, height, and width of the input feature map, respectively, and r denotes the channel down-sampling rate.

**Figure 7 sensors-24-05034-f007:**
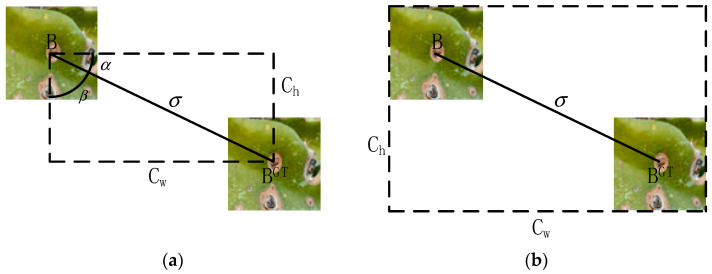
Schematic diagram of distance loss and angle loss calculation. (**a**) Angle cost. (**b**) Distance cost.

**Figure 8 sensors-24-05034-f008:**
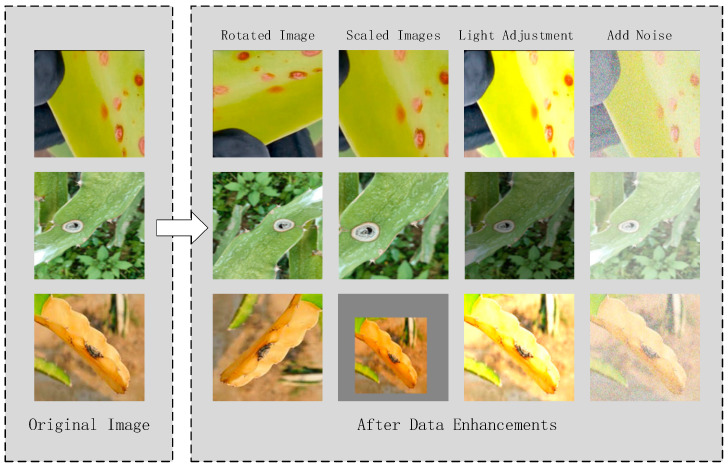
Comparison between the original plot and the enhanced data.

**Figure 9 sensors-24-05034-f009:**
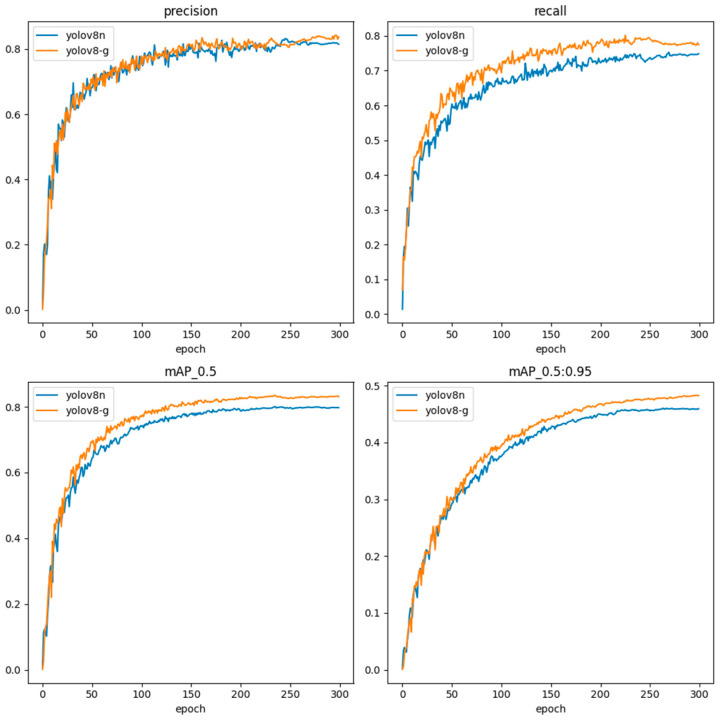
Visualization of results before and after YOLOv8n improvement.

**Figure 10 sensors-24-05034-f010:**
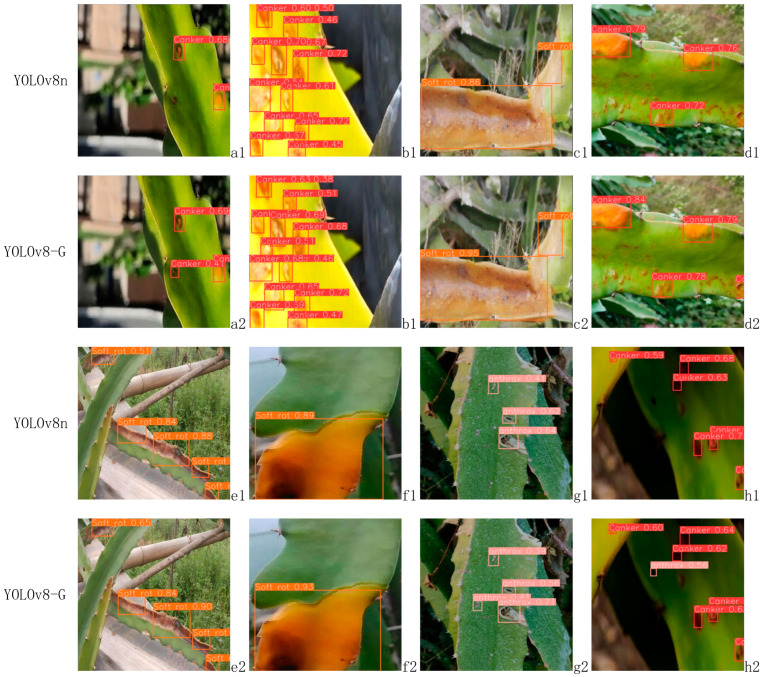
Comparison of the detection effect before and after YOLOv8n improvement.

**Figure 11 sensors-24-05034-f011:**
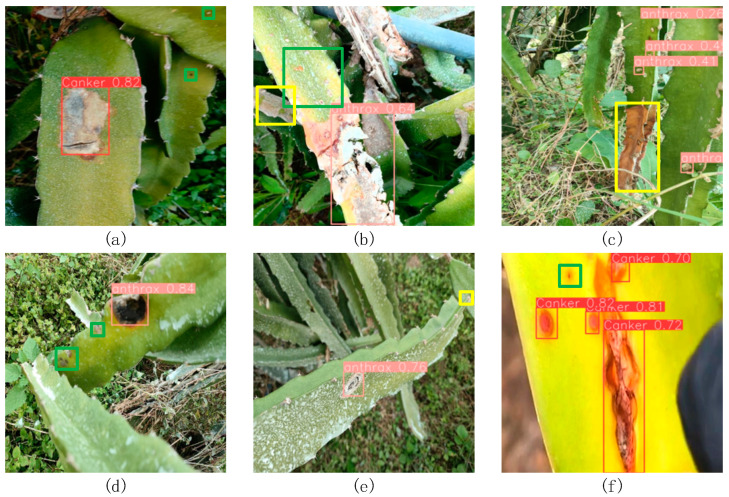
Schematic diagram of missed disease detection. (**a**,**d**,**f**) illustrate the model’s missed detections for tiny targets; (**c**,**e**) highlight missed detections in occluded regions; (**b**) demonstrates the model’s missed detections for both tiny targets and occluded regions. Note: The green boxes represent small target missed detections and the yellow boxes represent missed detections in the occluded area.

**Table 1 sensors-24-05034-t001:** Main characteristics and photos of dragon fruit stem diseases.

Types	Characteristic	Photo
Canker	During the initial stages of the illness, the lesions appear as round, light-colored sunken spots. In the middle stages, reddish-brown pinpoints appear in the center of the lesions, and eventually the pinpoints will expand over time to form brown raised spots.	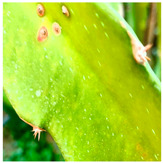
Anthrax	At the early stage of infection, brown lesions are produced on the stem, with reddish-brown round to irregularly shaped spots on the stem, slightly raised. At a later stage, the spots become larger, gradually connected into pieces, turning yellow with small black spots protruding on the surface of the stem.	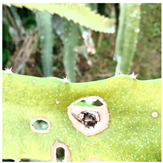
Soft rot	The lesions initially appear infiltrated and translucent, followed by a watery change in the lesion, which becomes slimy and softly rotting and emits a fishy odor, eventually spreading to the entire stem node.	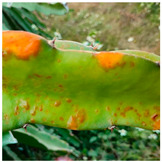

**Table 2 sensors-24-05034-t002:** Summary of dragon fruit stem disease data.

Types	Canker (pcs)	Anthrax (pcs)	Soft Rot (pcs)	Total (pcs)
Original images	242	265	231	738
Data-enhanced image	1000	1000	1000	3000

**Table 3 sensors-24-05034-t003:** Test environment configuration.

Configuration Name	Environmental Parameters
Operating system	Windows 10
CPU	Intel Core i5-12490F @ 3.00 GHz
GPU	NVIDIA GeForce RTX 4060ti (8 G)
Memory	16 G
Python	3.8
PyTorch	1.12
CUDA	11.6

**Table 4 sensors-24-05034-t004:** The configuration of experimental hyperparameters.

Batch Size	Epoch	Ir	Irf	Momentum	Weight_decay	Imgsz	Optimizer
32	300	0.01	0.01	0.937	0.0005	640	SGD

**Table 5 sensors-24-05034-t005:** Analysis of YOLOv8’s and YOLOv8-G’s performance.

Model	P (%)	R (%)	mAP50 (%)	mAP50:95 (%)	F1 Score (%)	Model Size (MB)	FLOPS (G)	FPS
YOLOv8n	83.2	75.2	79.8	46.0	78.0	6	8.1	86.5
YOLOv8-G	84.2	80.2	83.1	48.3	80.0	4.9	6.9	87.3

**Table 6 sensors-24-05034-t006:** Results of different model tests.

Model	P (%)	R (%)	mAP50 (%)	mAP50:95 (%)	F1 Score (%)	Model Size (MB)	FLOPS (G)	FPS
Faster-RCNN	45.1	63.6	55.8	30.5	52.3	108	273.4	2.6
SSD(VGG)	85.9	65.8	78.1	41.7	74.3	91.6	61.2	27.4
YOLOX-tiny	78.5	70.1	78.6	43.7	75.7	19.4	6.4	68.2
YOLOv8n	83.2	75.2	79.8	46.0	78.0	6.0	8.1	86.5
YOLOv9-t	79.7	77.8	81.8	48.5	79.0	6.1	11.0	90.9
YOLOv8-G	84.2	80.2	83.1	48.3	80.0	4.9	6.9	87.3

**Table 7 sensors-24-05034-t007:** Comparative test data of attention mechanisms.

Model	mAP50 (%)	mAP50:95 (%)	Parameters (M)	FLOPS (G)	FPS
YOLOv8n	79.8	46.0	31.51904	8.7	86.5
YOLOv8n-CBAM	80.9	46.3	32.17794	8.7	84.6
YOLOv8n-SimAM	81.4	46.9	31.51904	8.7	83.1
YOLOv8n-SE	81.9	46.1	31.60096	8.7	84.4
YOLOv8n-CA	82.1	47.2	31.58584	8.7	83.8

**Table 8 sensors-24-05034-t008:** Loss function comparison test data.

Model	mAP50(%)		Parameters (M)	FLOPS (G)
	IoU	Focal + IoU		
YOLOv8n + CIoU	79.8	80.5	30.11417	8.2
YOLOv8n + DIoU	80.9	81.0	30.11433	8.2
YOLOv8n + EIoU	81.4	81.4	30.11428	8.2
YOLOv8n + GIoU	82.1	82.2	30.11437	8.2
YOLOv8n + SIoU	82.2	82.4	30.11430	8.2

**Table 9 sensors-24-05034-t009:** Ablation test results.

Test	C2f-Faster	CA	CARAFE	SIoU	P (%)	R (%)	mAP50 (%)	mAP50:95 (%)	F1-Score (%)	Model Size (MB)	FLOPS (G)	FPS
1					83.2	75.2	79.8	46.0	78.0	6	8.1	86.5
2	✔				83.1	76.5	80.5	46.5	79.0	4.7	6.5	83.1
3	✔	✔			83.6	78.3	81.0	47.2	79.0	4.7	6.5	82.3
4	✔	✔	✔		83.9	78.6	82.3	48.1	80.0	4.9	6.9	92.1
5	✔	✔	✔	✔	84.2	80.2	83.1	48.3	80.0	4.9	6.9	87.3

## Data Availability

Data are contained within the article.

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
