# Peer review of "YOLOv8-G: An Improved YOLOv8 Model for Major Disease Detection in Dragon Fruit Stems"

_sensors, 2024, doi:10.3390/s24155034_

Round 1
Reviewer 1 Report
Comments and Suggestions for Authors
The work could be interesting. More robustness discussion and comparison are expected. Further improvement is required.
1. More research background and discussion are required e.g. Xie, Y., Cai, J., Bhojwani, R., Shekhar, S., & Knight, J. (2019). A locally-constrained YOLO framework for detecting small and densely-distributed building footprints. International Journal of Geographical Information Science, 34(4), 777–801. https://doi.org/10.1080/13658816.2019.1624761.
Boudjit, K., & Ramzan, N. (2021). Human detection based on deep learning YOLO-v2 for real-time UAV applications. Journal of Experimental & Theoretical Artificial Intelligence, 34(3), 527–544. https://doi.org/10.1080/0952813X.2021.1907793.
2. More robustness studies including feature extraction and capture condition should be discussed.
Comments on the Quality of English LanguageNA
Reviewer 2 Report
Comments and Suggestions for Authors
The article presents a set of improvements for the well-known YOLOv8 model to detect diseases in Dragon Fruit Stems.
There are several issues in the article that authors have to correct.
First, the introduction is more focused on the related work, giving very little space to the motivation of the problem. Similarly, the contributions of the paper are found briefly in the last paragraph.
Second, the related work section is more about the model description and the data preparation than the related work itself, which has been pushed to the introduction.
In general figures explaining the model components are fine. However, it would be helpful to include references in the text that focus on specific components and parts of the figures. As an example, the description of Figure 7 provides some numbering, however, these numbers can't be find in the Figure.
The equation numbers on page 10 are incorrect.
At some point in the text it is said that a hyperparameter has been obtained by using evolutionary algorithms. Later there is a table showing the set of hyperparameters, but there is no clue on how they have been obtained. Further, it is unclear if the results for all models use the same set of hyperparameters and how these are tight to the model that has been used to obtain them. In other words, are the results consistent if the hyperparameters are obtained by optimizing other models as well?
There is a small typo in Table 3, I guess it is Pytorch, not Pytorth.
It is unclear if the results presented are from the validation set or the test set. This is confusing when looking at Table 5, which one would assume refers to the test set results, but then finding Figure 9, which seems to refer to the validation set.
Figures in Table 6 do not clearly show the improvement of the model. The sizes of the boxes are not apparently smaller and the reviewer has only found 1 small extra box provided by the YOLOv8-G version.
Last but not least, the article motivates the usage of a lightweight version of the YOLO model for its applicability at the Edge. However, the paper does not present any comparative analysis of the models used in Edge devices, finishing the paper with a hypothesis that has not been tested. (Simply put, how much better is the performance of the new model in an Edge device compared to the others)
Comments on the Quality of English LanguageEnglish is fine, just review for typos.
Reviewer 3 Report
Comments and Suggestions for Authors
Abstract
• The line 10 “... dragon fruit, in order” should be separated by "." instead of ",”.
• Line 13-17 could be shorter without the need to explain too much detail there.
• The purpose of the research is not clear enough. Need to make it clear why it needs to use YOLO to do the detection.
Introduction
• It needs to have a more detailed explanation about relationship between the stem disease and fruit itself by providing a stronger argument with support of rich data.
• It needs to address what is the motivation of using deep learning approaches instead of other approaches to detect dragon fruit stem disease.
• The sentence on line 40-41 needs to be corrected. The sentence is not clear enough, suddenly it explains the 2 categories of target detection algorithm after explaining AI technology in agriculture.
• The sentences in line 48 to line 51 are not well connected. The author should not just directly explain other research work without any context.
• Too much space (line 56-83) used only to introduce the YOLO-series, which is not needed for the Introduction section.
• Same as before, in line 87 to 91, the author should not just directly explain other research work without any context.
• Is there any other recent research related to dragon fruit stem disease detection using deep-learning approaches? If there are any, the author should introduce it more in the Introduction section. If it is not, the author can focus on introducing other kinds of plant disease using deep-learning approaches, especially using YOLO-series.
• Need to address which part of YOLOv8 that has been modified by the author.
Related Work
• This section should be written about more detailed previous research related to deep learning approaches to detect dragon fruit stem disease.
• Dataset preparation and Dataset pre-processing should be moved into
Experimental section to describe what is the dataset that the author used to do this experiment.
• Need to have clearer and high-resolution images on Table 1.
Methodology and Design
• The sentences in lines 222-224 “The construction ....” is not related to the previous and next sentences.
• Figure 5 should be added by the original structure C2f that used in original YOLOv8 so the reader can see the difference between them.
• The sentence in line 240-241 “The disease detection ...” is not well related with previous and the next sentence.
• The citation about CARAFE upsampling should be placed in the beginning, before explaining the details of it.
• The equations 8-11 are not well explained.
• Needs to explain what is SIoU first before explaining the formula.
Experimental Results and Analysis
• The images quality on Table 6 should be better, hard to read the confidence value of the detection
• Table 6 should be included with a case when the original YOLOv8n are not able to detect the disease, while the proposed model can easily detect it
• Should use another recent deep learning algorithm to compare with the proposed model
• In line 425, it said that YOLOv8n outperforms other detection models, but in the end, it said that the proposed model has higher performance
• What is the reason to put only the mAP50 to comparing different loss function?
Discussion
• It is ambiguous that the whole sentences are a discussion or a conclusion.
• The sentences in lines 523-524 “The improved model can effectively adjust to varying illumination conditions” does not have a strong basis stated in the paper to prove it.
Other
• Need to more specifically said that the research is about dragon fruit stem disease, not the fruit itself. Some of the sentences in the paper did not mention it.
• No conclusion provided
• It should be better if author provide some sample of images with the explanation to describe the challenge of detecting dragon fruit stem disease.
Moderate editing of English language required.
Reviewer 4 Report
Comments and Suggestions for Authors
The paper presents a clear and concise objective of developing an improved YOLOv8 model for disease detection in dragon fruit stems. To further improve the quality of the paper, the following issues should be addressed.
1. The introduction of the C2f-faster module and the CARAFE up-sampling operator demonstrates innovation. However, the paper could provide more insight into why these particular components were chosen over other existing technologies in the field.
2. The literature review is comprehensive but could benefit from a more critical analysis of how the YOLOv8-G model compares to state-of-the-art methods beyond the YOLO series.
3. The network architecture diagrams are informative but could be enhanced with additional annotations or a legend for better clarity, especially for readers unfamiliar with YOLOv8's structure.
4. The training process could be described in more detail, including the division of training and validation sets, the choice of optimizer, and any hyperparameter tuning strategies employed.
5. While mAP is a standard metric, the paper could consider additional metrics such as F1-score or IoU to provide a more comprehensive evaluation of the model's performance.
6. The comparative analysis with other models is limited to performance metrics. A cost-benefit analysis or a discussion on the trade-offs between accuracy and computational efficiency would be insightful.
7. The paper introduces a coordinate attention mechanism but does not deeply explore its impact on various types of dragon fruit stem diseases individually.
8. The paper replaces the CIoU with the SIoU loss function but does not thoroughly explain the benefits this change brings to the model's performance.
9. There is a lack of discussion on the interpretability of the model. Understanding which features the model relies on to make predictions would be valuable.
10. The paper concludes with a brief discussion of potential future work. Expanding on this section to include specific areas for improvement or additional features to be incorporated would be beneficial.
Round 2
Reviewer 1 Report
Comments and Suggestions for Authors
The improvement is reasonably good. More critical discussion on different applications could be explored. e.g. Müller, A. M., & Hausotte, T. (2022). Improving template-based CT data evaluation by integrating CMM reference data into a CAD model-based high fidelity triangle mesh. Nondestructive Testing and Evaluation, 37(5), 692–706. https://doi.org/10.1080/10589759.2022.2091135.
Comments on the Quality of English LanguageOK
Author Response
Please see the attachment.
The revised statements are highlighted in blue within the article.

Reviewer 3 Report
Comments and Suggestions for Authors
This revision has addressed most of the reviewer's concerns and can be accepted with minor English editing.
Comments on the Quality of English LanguageMinor edit is recommended.
Author Response

(The authors gave the same response as above.)

Reviewer 4 Report
Comments and Suggestions for Authors
All questions answered and revised by the author
Author Response

(The authors gave the same response as above.)
